# Chloroquine-Induced DNA Damage Synergizes with Nonhomologous End Joining Inhibition to Cause Ovarian Cancer Cell Cytotoxicity

**DOI:** 10.3390/ijms23147518

**Published:** 2022-07-07

**Authors:** María Ovejero-Sánchez, Jorge Rubio-Heras, María del Carmen Vicente de la Peña, Laura San-Segundo, Jesús Pérez-Losada, Rogelio González-Sarmiento, Ana Belén Herrero

**Affiliations:** 1Institute of Biomedical Research of Salamanca (IBSAL), 37007 Salamanca, Spain; maria.os@usal.es (M.O.-S.); duckey@usal.es (L.S.-S.); jperezlosada@usal.es (J.P.-L.); 2Molecular Medicine Unit, Department of Medicine, University of Salamanca, 37007 Salamanca, Spain; jorrubher@usal.es (J.R.-H.); carmenv99@usal.es (M.d.C.V.d.l.P.); 3Institute of Molecular and Cellular Biology of Cancer (IBMCC), University of Salamanca-CSIC, 37007 Salamanca, Spain

**Keywords:** chloroquine, panobinostat, DNA damage and repair, nonhomologous end joining, KU-57788, NU-7026, SCR7

## Abstract

Ovarian cancer (OC) is the most lethal gynecological malignancy; therefore, more effective treatments are urgently needed. We recently reported that chloroquine (CQ) increased reactive oxygen species (ROS) in OC cell lines (OCCLs), causing DNA double-strand breaks (DSBs). Here, we analyzed whether these lesions are repaired by nonhomologous end joining (NHEJ), one of the main pathways involved in DSB repair, and if the combination of CQ with NHEJ inhibitors (NHEJi) could be effective against OC. We found that NHEJ inhibition increased the persistence of γH2AX foci after CQ-induced DNA damage, revealing an essential role of this pathway in the repair of the lesions. NHEJi decreased the proliferation of OCCLs and a strong in vitro synergistic effect on apoptosis induction was observed when combined with CQ. This effect was largely abolished by the antioxidant N-Acetyl-L-cysteine, revealing the critical role of ROS and DSB generation in CQ/NHEJi-induced lethality. We also found that the NHEJ efficiency in OCCLs was not affected by treatment with Panobinostat, a pan-histone deacetylase inhibitor that also synergizes with CQ in OCCLs by impairing homologous recombination. Accordingly, the triple combination of CQ-NHEJi-Panobinostat exerted a stronger in vitro synergistic effect. Altogether, our data suggest that the combination of these drugs could represent new therapeutic strategies against OC.

## 1. Introduction

Ovarian cancer (OC) is the second most common gynecologic neoplasia and it has the highest mortality rate among them, representing the fifth leading cause of cancer death among women [1]. This high mortality is mainly due to diagnoses at advanced stages of the disease (stages III-IV) when the tumor has already spread to the peritoneal cavity and other organs [2]. Moreover, disease relapse is quite common after primary cytoreductive surgery and a standard platinum/taxane-based chemotherapy [3,4], and after further treatments with different chemotherapeutic agents, such as platinum-derived drugs, antiangiogenic drugs, or PARP inhibitors [5,6,7]. Therefore, there is a clear need to develop more therapeutic strategies, such as new drug combinations to overcome resistance and improve ovarian cancer survival.

Chloroquine (CQ), a well-known autophagy inhibitor, is a 4-aminoquinoline-based drug initially discovered and used to prevent and treat malaria [8]. This compound has also been used as an anti-inflammatory agent to treat several inflammatory diseases such as rheumatoid arthritis or lupus erythematosus [8,9]. In addition, CQ is considered a potential anticancer agent; it has been described to reduce hypoxia, cancer cell invasion, and metastasis [10]. Moreover, several preclinical results and clinical trials have shown that this compound, in combination with other antitumor drugs, increases the cytotoxic effect and sensitizes tumor cells to radiotherapy or chemotherapy [9,11,12,13,14,15,16,17,18,19,20,21,22,23]. The effect of CQ as an anticancer drug has been mainly attributed to autophagy inhibition [9,12,13,14,21,22,23], a process that promotes cancer cell survival by recycling intracellular organelles and proteins [11,24,25,26], and it is considered a resistance mechanism against chemotherapy [27]. However, it has also been reported that the ability of CQ to inhibit autophagy is not the only mechanism by which it exerts its antitumor effect [10,15,28,29,30,31]. In this regard, we have recently published that CQ increases reactive oxygen species (ROS) in OC cell lines, causing DNA double-strand breaks (DSBs) [32], the most lethal form of DNA damage [33]. In that work, we also described that the combination of CQ with Panobinostat (LBH) exerted a strong synergistic effect, which was explained by the ability of this histone deacetylase inhibitor (HDACi) to reduce the efficiency of homologous recombination (HR), one of the pathways involved in DSB repair. Some reports have found that HDACis may also affect the efficiency of nonhomologous end joining (NHEJ) [34,35], the other primary mechanism involved in DSB repair and, therefore, another potential candidate for the repair of oxidative damage [36]. However, although some HDACis seem to decrease the amount of some proteins involved in the pathway in certain types of tumor cell lines (i.e., Ku70, Ku80, and DNA-PKcs) [37,38,39], contradictory results have been published on whether these compounds decrease or increase the frequency of NHEJ at the DSB sites [34,35]. Functional assays to measure the efficiency of NHEJ in the presence of Panobinostat in OC cells have not been carried out, nor has the impact of NHEJ inhibition on the repair of CQ-induced DSBs been explored.

In this study, we first investigated whether the HDACi Panobinostat affects the efficiency of NHEJ using a reporter assay based on green fluorescent protein reconstitution. We found that this compound did not alter NHEJ efficiency, which prompted us to explore the role of this pathway in the repairing of CQ-induced DSBs, and also to evaluate the effect of triple combinations, CQ-NHEJi-Panobinostat. We describe that the NHEJ pathway is essential for the repair of DSBs caused by CQ, and, importantly, that the combination of CQ with different NHEJ inhibitors exerts a potent synergistic effect in OC cell lines, representative of the most common histopathological OC subtypes. This effect, which depends on CQ-induced ROS production, was even potentiated by the triple combination CQ-NHEJi-Panobinostat. Taken together, our results, based on in vitro approaches, open the possibility of exploring these new combination strategies in clinical studies.

## 2. Results

### 2.1. Panobinostat Does Not Affect NHEJ Efficiency in OCCLs

It has been described that some HDAC inhibitors, including Panobinostat, affect DNA DSB repair by inhibiting homologous recombination (HR) [32,40,41,42]. Indeed, we have recently reported that LBH inhibits the correct recruitment of Rad51 to DSB sites in OCCLs, which explains the HR defect [32]. However, the effect of HDAC inhibitors on DSB repair by NHEJ seems to be controversial [34,35]. We chose Panobinostat for several reasons: first, because it is structurally similar to vorinostat while exhibiting a higher potency [43]; second, because Panobinostat has proven its efficacy in vitro and in vivo in ovarian cancer [44], which might be related to the fact that most epithelial ovarian carcinomas show epigenetic changes and overexpression of histone deacetylases (HDACs) [45]; and third, because it decreases HR repair and approximately half of all women diagnosed with ovarian cancer are HR-proficient. In this regard, a recent study indicates that Panobinostat in combination with olaparib reduced peritoneal metastases and tumor burden in a syngeneic mouse model of ovarian cancer [46].

To test whether Panobinostat affected the NHEJ pathway in OCCLs, we first analyzed the levels of some proteins involved in this repair mechanism in the SK-OV-3 cell line at different times post-treatment. We found that proteins levels were similar in all the samples analyzed (Figure 1A, left panel), and also in the other OCCLs (Figure 1A, right panel). However, to further explore a putative effect on NHEJ efficiency, we next used an extrachromosomal functional assay where end joining is determined by measuring the cell’s ability to recircularize an enzyme-digested plasmid (pEGFP-Pem1-Ad2) (Figure 1B). Plasmid recircularization results in the formation of the green fluorescent protein (GFP), and GFP+ cells can be easily detected and quantified by flow cytometry. SK-OV-3 and IGROV-1 cell lines were pre-treated or not with Panobinostat or NU-7026, a well-known NHEJ inhibitor, for 24 h, and then transfected with linearized pEGFP-Pem1-Ad2 or supercoiled pEGFP-Pem1 together with psDsRedN1 to correct for transfection efficiencies. Cells were then incubated again with LBH or NU-7026 for 72 h. The number of GFP+ cells obtained by transformation with the linear *HindIII*- or *SceI*-digested plasmids was similar in the LBH-treated and untreated samples, giving rise to a similar % of NHEJ, which indicates that this HDACi does not alter DSB repair by NHEJ (Figure 1C). On the contrary, NU-7026-treated cells decreased the end-joining efficiency, as expected. LBH-doses used in these assays were effective in SK-OV-3 and IGROV-1 cell lines, causing a 50% reduction in cell survival. To confirm these data obtained using episomal plasmids, we used an intrachromosomal substrate, NHEJ-C, that was integrated into the chromatin of IGROV-1 and SK-OV-3 cell lines. DSBs were generated by the transfection of the stable cell lines with an *I-SceI* endonuclease-expressing plasmid, and the NHEJ efficiency was estimated 72 h later as the ratio of EGFP+/DsRed+ cells. Although treatment with LBH decreased cell survival (data not shown), no significant differences in the number of live NHEJ-proficient cells were found in samples that had been treated with LBH compared with untreated cells, which confirmed that LBH did not alter DNA repair by NHEJ in OCCLs (Figure 1D).

### 2.2. Chloroquine Induces DSBs, Which Are Repaired by NHEJ

CQ has been demonstrated to exert pleiotropic cellular effects, including the generation of reactive oxygen species (ROS) [32,49,50,51,52]. We have recently shown that CQ-induced ROS production in OCCLs leads to the generation of DNA-DSBs [32]. These types of lesions are usually repaired by two main pathways, HR and NHEJ; the latter has been described to occur with a higher frequency and to be faster than HR [48]. Based on these data, we decided to study whether CQ-induced DSBs could be repaired by NHEJ. SK-OV-3 and OVCAR-8 OCCLs were treated or not with CQ for 24 h, the drug was then removed, and the cells were cultured in the presence or the absence of three different NHEJ inhibitors: KU-57788, NU-7026, and SCS7 pyrazine (Figure 2A). KU-57788 and NU-7026 inhibit DNA-PK, an essential protein complex that facilitates the synapsis of broken DNA ends [53,54], whereas SCR7 pyrazine inhibits DNA ligase IV, the enzyme that seals the DNA ends. After 48 h in the presence or absence of any of the three inhibitors, cells were collected, and the phosphorylation of H2AX (γH2AX), a marker of DSBs, was analyzed under all the conditions by immunofluorescence (Figure 2B). We found that CQ was able to induce DSBs in both OCCLs analyzed (quantifications are shown in Figure 2C). However, the lesions disappeared after 48 h of drug removal. On the contrary, the DNA damage remained when cells were exposed to NHEJ inhibitors upon CQ removal. Treatment with NHEJi alone barely affected the percentage of cells with γH2AX foci found in untreated samples (Figure 2B,C). These data clearly indicated that CQ-induced DSBs are, at least in part, repaired by the NHEJ pathway.

### 2.3. NHEJ Inhibitors Decrease Proliferation Rates and Induce Apoptosis in OCCLs

We have previously reported that treatment with CQ decreased cell proliferation in OCCLs [32]. To determine whether NHEJ inhibitors affect the growth of these cells, we performed MTT assays using different concentrations of KU-57788, NU-7026, and SCR7 pyrazine and several incubation times. We found that the three compounds inhibited cell proliferation in a dose- and time-dependent manner in all cell lines analyzed (Figure 3A and Appendix A). However, KU-57788 and NU-7026 were more potent than SCR7 pyrazine in inhibiting cell proliferation and exerted a rapid effect that was already observed 24 h post-treatment. Both drugs presented IC50 values that ranged from 0.96 µM to 11.07 µM, while IC50 values of SCR7 pyrazine ranged from 204.5 µM to 329.8 µM (Table 1).

To further characterize the antiproliferative activity of the NHEJ inhibitors, a cell cycle study was carried out. We found that KU-57788 increased the number of cells in the G1 phase compared with the control in all cell lines tested (Appendix A), in agreement with previous reports [55,56,57,58]. This increase was accompanied by a concomitant decrease in G2/M. The same effects were also found in A2780 and IGROV-1 treated with NU-7026 (Appendix A), but were not observed in OVCAR-8 and SK-OV-3 at the doses and conditions employed. The sub-G0 population was also quantified after treatment with these drugs. A statistically significant increase was found in all cell lines analyzed (*p*-value < 0.05) (Appendix A), indicating that KU-57788 and NU-7026 not only inhibit cell proliferation but also produce a cytotoxic effect. The cytotoxicity to NU-7026 in OVCAR-8 and SK-OV-3 was lower than that observed for A2780 and IGROV-1, which is consistent with the absence of G1 arrest in those cell lines. Treatment with SCR7 hardly affected cell cycle profiles in all the cell lines tested (Appendix A).

Next, we analyzed whether the cell death caused by KU-57788 and NU-7026 was due to the induction of apoptosis. Annexin V/PI staining revealed that both compounds reduced cell viability (Annexin V-/PI-) in OCCLs in a dose-dependent manner (Figure 3B). However, KU-57788 required lower doses than NU-7026 to induce apoptosis.

### 2.4. Combination of Chloroquine and NHEJ Inhibitors Synergistically Induces Cell Death in OCCLs

Once we established that CQ-induced DNA damage required the NHEJ pathway to be repaired, we analyzed the effect of combining this drug with different NHEJ inhibitors on apoptosis. As shown in Figure 4, Appendix A, the percentage of live cells was much lower in the combined treatments than in monotherapy. These results were also verified by clonogenic assays (Appendix A). To determine the interaction between CQ and these inhibitors, the combination indices at the different drug doses were calculated using Compusyn software (version 1.0). In all cases, CIs were below 1, indicating synergistic interactions.

### 2.5. Cell Death Induced by the Combination of CQ and NHEJ Inhibitors Depends on ROS Production

We previously described that CQ-induced DSBs and cell death caused by CQ/LBH combination depended on ROS production, as adding the ROS scavenger N-acetylcysteine exerted a protective effect [32]. To analyze whether ROS generation was also crucial in triggering apoptotic cell death by the combination CQ-NHEJi, cells were treated with the different drugs in the presence or absence of N-acetylcysteine. Cell survival was analyzed after 72 h of incubation by Annexin/PI staining. As shown in Figure 5, adding the antioxidant significantly prevented cell death caused by the combination of CQ and NHEJi.

### 2.6. The Triple Combination CQ-NHEJi-LBH Exerts a Strong Synergy and Higher Efficacy Compared with Double Combinations

Considering that the double combinations CQ/LBH and CQ-NHEJi exerted a synergistic effect on OC cell survival and that LBH inhibited HR but did not affect the efficiency of DSB repair by NHEJ, we hypothesized that the triple combination might synergistically increase OC cell death. To test this hypothesis, A2780, IGROV-1, and SK-OV-3 cell lines were treated with CQ, LBH, and NU-7026/KU-57788 at constant ratios, and the percentage of apoptotic cells was measured. As shown in Figure 6, the triple combination CQ-LBH-NHEJi exerted a strong synergistic effect that was higher than double combinations, especially when NU-7026 was employed, with the exception of the SK-OV-3 cell line treated with CQ-LBH-KU-57788.

## 3. Discussion

CQ, an antimalarial and anti-inflammatory drug, is one of the most prominent cases of drug repurposing in cancer [59]. Several reports and clinical trials have revealed the benefit of combining this compound with different chemotherapeutic drugs [9,11,12,13,14,16,17,18,19,20,21,22,23]. In this study, we describe a synergistic effect between CQ and NHEJ inhibitors for the first time. We show that CQ-induced DSBs are repaired by NHEJ, and avoidance of this pathway, using different inhibitors, induces a robust synergistic effect that could be therapeutically exploited.

CQ is a weak base that prevents autophagy by increasing the lysosomal pH and impairing autophagosome fusion with lysosomes [9,60]. Inhibition of autophagy might explain the synergistic effects of CQ when combined with chemo- and radiotherapy. However, several reports have found that the ability of CQ to block autophagy is not the only mechanism that explains its antitumor effect [10,15,28,29,30,31]. Thus, CQ could efficiently reduce drug degradation and reverse P-gp-induced drug sequestration to lysosomes by increasing lysosomal pH [15]. Additionally, it could buffer the tumor milieu allowing the entry of basic chemotherapeutic drugs into tumor cells. CQ also affects inflammatory responses [61], and reduces hypoxia, cancer cell invasion, and metastasis [10]. It has also been reported that CQ could eliminate cancer stem cells via an epigenetic mechanism, by altering DNA methylation [30]. In addition, both our group and others have reported that CQ increases oxidative stress in cancer cells [21,49,50,51,52,62,63]. This effect may either be mediated by autophagy inhibition that would avoid the elimination of damaged mitochondria, the primary source of ROS [64], or by CQ-induced mitochondrial cristae damage, as was recently found by Liang et al., using electron microscopy [65]. Cristae damage leads to mitochondrial membrane depolarization, a reduction in cytochrome c oxidase activity, and an accumulation of superoxide.

Nevertheless, CQ-induced oxidative stress could drive oxidative pressure to toxic levels selectively in tumor cells that are known to exhibit high endogenous oxidative levels [65,66]. However, intact or even increased DNA repair levels in cancer cells [67] may protect them from oxidative damage, preventing the accumulation of toxic DNA lesions. In fact, we previously found that Panobinostat synergized with CQ in OCCLs by reducing HR efficiency [32], one of the two main pathways involved in DSBs repair. The other pathway, NHEJ, occurs more often, is faster than HR [48], and is also supposed to deal with oxidative DNA lesions [36]. Here, we confirmed the essential role of NHEJ in CQ-induced DSB repair; we found that inhibition of this pathway, using three different NHEJ inhibitors, increased the persistence of γH2AX foci after drug removal. Moreover, we show that CQ synergizes with the three inhibitors in all four OCCLs analyzed, which represents the most common histopathological OC subtypes.

We previously reported that CQ-induced DSBs and cell death caused by CQ/LBH combination were primarily abolished by the ROS scavenger N-acetylcysteine. Here, we found that the addition of this antioxidant also avoided CQ/NHEJi-induced cell apoptosis. Together, these results reveal the critical role of ROS production and subsequent DSB generation in the lethality produced by the mentioned combinations. We found that inhibition of both DNA repair pathways, NHEJ and HR, highly increased the synergistic effect with CQ, except in the SK-OV-3 cell line. This cell line bears a *TP53* homozygous mutation (Expasy, Cellosarous), whereas the A2780 cell line is wild-type for *TP53* and the IGROV-1 cell line has a heterozygous mutation (c.377A>G, Expasy, Cellosarous) classified as an uncertain significance variant (ClinVar = VCV000458541). Therefore, p53 loss in SK-OV-3 might avoid apoptosis induction even in the absence of both DNA repair pathways. In this regard, prior studies have demonstrated that p53-deficient cells have lost the balance between HR activation and repression [68] and also that DNA-PK inhibition leads to the use of an alternative end-joining mechanism to promote cell viability in response to DSBs [69]. Abrogation of both pathways might potentiate alternative mechanisms promoting cell survival; however, testing this hypothesis would require further research.

The ability of the NHEJ inhibitors used in this work in combination with several genotoxic drugs to avoid tumor growth has been observed in different cancer types. Thus, it has been described that NU-7026, a DNA-PK inhibitor, potentiates the cytotoxicity of topoisomerase II poisons or doxorubicin used in the treatment of leukemia [70], synergizes with Irinotecan in colon cancer cells [71], and sensitizes cancer cells to ionizing radiation [72]. A radio- and chemo-sensitization effect has also been reported for KU-57788, another potent and selective DNA-PK inhibitor [55,57,73,74], which justified further development of DNA-PK inhibitors for clinical use. In OC, elevated DNA-PK expression has been associated with poor cancer-specific survival [75] and combinations of genotoxic treatments with DNA-PK inhibitors are being tested [76,77]. Finally, SCR7 and its derivatives, which inhibit DNA Ligase IV, the enzyme responsible for sealing double-strand breaks (DSBs), have been shown to reduce tumor progression in mouse models, especially when co-administered with DSB-inducing drugs [78]. In this work, we found that KU-57788 and NU-7026 were more potent than SCR7 in OC cell lines. KU-57788 and NU-7026 inhibit DNA-PK activity, whereas SCR7 inhibits DNA ligase IV. Higher doses of DNA ligase IV inhibitor might be necessary to achieve the same effect obtained with DNA-PK inhibitors. In fact, SCR7 IC50 reported by the supplier (MedChemExpress, South Brunswick Township, NJ, USA) is higher than those reported for the DNA-PK inhibitors. It has also been described that OC cells required higher doses of SCR7 than other cancer cell lines [78]. The authors found that IC50s for SCR7 differed between the cancer cell lines analyzed, with A2780 being one of the most resistant OC cell lines, in agreement with our results [78]. In addition, treatment with SCR7 in an OC xenograft model resulted in tumor regression with no obvious adverse effects [78].

It is known that cells with high rates of proliferation are more susceptible to DNA damage and exhibit higher levels of reactive oxygen species [65,66,79]. The latter can result from increased metabolic activity, as ATP synthesis produces ROS during normal oxygen metabolism [80]. Several works have combined genotoxic agents with DNA repair inhibitors in vivo and have found cytotoxic effects in tumor cells and lower toxicity in normal cells [76,77,81]. Therefore, CQ in combination with DNA repair inhibitors could drive DNA damage and oxidative pressure to toxic levels selectively or preferentially in tumor cells. The advantages of using CQ together with NHEJ inhibitors and/or HR inhibitors need to be evaluated in in vivo experiments. In this regard, although KU-57788 has been shown to provide excellent proof-of-principle in vitro and in vivo chemo- and radio-sensitization, it is known to present a limited aqueous solubility, which restricts its further development [74]. Additionally, CQ, with proven efficiency in preclinical studies, has been reported to present difficulties in passing through the cell membrane in the presence of acidic extracellular microenvironments, which is characteristic of tumors [9,11]. This problem could be overcome by the encapsulation of CQ using nanoparticles, an approach that has been demonstrated to increase not only the CQ antitumor effect but also the efficient delivery of drugs to tumor sites with superior pharmacokinetic profiles, prolonged drug circulation, and reduced systemic toxicity in vivo [15,16,82,83,84]. The encapsulation of CQ together with an NHEJ inhibitor could then be explored for the treatment of OC to recapitulate the efficacy of this combination shown in this work.

## 4. Materials and Methods

### 4.1. Cell Lines and Culture Conditions

The ovarian cancer cell lines (OCCLs) A2780, IGROV-1, OVCAR-8, and SK-OV-3 were used in this work. OVCAR-8 and SK-OV-3 were acquired from the American Type Culture Collection (ATCC), A2780 from the European Collection of Authenticated Cell Cultures (ECACC), and IGROV-1 from Merck Millipore, Burlington, MA, USA. IGROV-1 and A2780 were cultured in RPMI 1640 medium (Gibco, Waltham, MA, USA) supplemented with 10% heat-inactivated fetal bovine serum (FBS) (Gibco) and 1% penicillin/streptomycin (Gibco). SK-OV-3 and OVCAR-8 were cultured in Dulbecco’s modified Eagle’s medium (DMEM) (Gibco) supplemented with 10% FBS and 1% penicillin/streptomycin. All cells were incubated at 37 °C in a 5% CO_2_ atmosphere. The presence of mycoplasma was routinely checked with the MycoAlert kit (Lonza, Basel, Switzerland), and only mycoplasma-free cells were used in the experiments.

### 4.2. Reagents

Chloroquine (CQ) and N-Acetyl-L-Cysteine were purchased from Sigma-Aldrich, St. Louis, MO, USA. Nonhomologous end joining inhibitors (NHEJi) KU-57788 (KU), NU-7026 (NU), and SCR7 pyrazine (SCR7), a stable form of SCR7, were obtained from MedChemExpress, South Brunswick Township, NJ, USA. Panobinostat (LBH) was provided by Novartis Pharmaceuticals, Basel, Switzerland.

### 4.3. Western Blot

Cells were treated or not with Panobinostat and then resuspended in lysis buffer (50 mM Tris-HCl pH 7.4, 150 mM NaCl, 1 mM EDTA, and 1% Triton X-100) containing protease inhibitors (Complete, Roche Applied Science, Indianapolis, IN, USA). Protein concentration was measured using the Bradford assay (BioRad, Hercules, CA, USA). Protein samples (30 μg/lane) were subjected to SDS-PAGE and transferred to PVDF membranes (Immobilon-PSQ PVDF Membrane, Merck Millipore, Burlington, MA, USA). After blocking, membranes were incubated with primary antibodies against the following proteins: Ku70 (1:1000, sc-5309, Santa Cruz Biotechnology, Dallas, TX, USA), Ku80 (1:2000, sc-56136, Santa Cruz Biotechnology), and β-actin (1:10000, A5441, Sigma-Aldrich). β-actin was used for the loading control. Goat Anti-Mouse IgG (H+L) DryLight^TM^ 680 Conjugated (1:10,000, 35518, Invitrogen, Waltham, MA, USA) was used as the secondary antibody. Immunoblots were incubated for 1 h at room temperature and developed using the Odyssey infrared imaging system (LI-COR Biosciences, Lincoln, NE, USA).

### 4.4. NHEJ Assays

To determine NHEJ levels, the end-joining reporter plasmid pEGFP-Pem1-Ad2 was used [85]. This plasmid was digested with *HindIII* or *I-SceI* enzymes to eliminate the Ad2 sequence within the Pem1 intron and generate compatible or incompatible ends. The EGFP signal can be restored when the plasmid is recircularized thanks to the NHEJ activity of the transfected cells. IGROV-1 and SK-OV-3 cell lines were pre-incubated with different concentrations of Panobinostat or NU-7026 for 24 h. Then, 10^6^ cells were co-transfected with 0.5 µg of either linearized pEGFP-Pem1-Ad2 or supercoiled pEGFP-Pem1 (the result of the religation of HindIII-digested pEGFP-Pem1-Ad2) together with 0.5 µg of pDsRed-N1 plasmid (Clontech, Mountain View, CA, USA) to evaluate the transfection efficiency. Transfections were performed using the 1SM buffer [86] and the Amaxa Nucleofector device (Lonza). The programs used were X-001 for IGROV-1 and V-005 for SK-OV-3. After transfection, cells were incubated again with LBH or NU for 72 h. Green (EGFP) and Red (DsRed) fluorescences were measured by flow cytometry using a BD Accuri™ C6 Plus Flow Cytometer. The NHEJ efficiency was calculated as the ratio of EGFP-positive cells from recircularized linear plasmid to the number of green cells from the transfections of pEGFP-Pem1 plasmid after normalizing according to the number of red cells.

### 4.5. Construction of Cell Lines for Detecting NHEJ Efficiency

SK-OV-3 and IGROV-1 cell lines were transfected with 0.5 μg of the NHEJ-C reporter construct linearized by digestion with *NheI* [48]. G418 was added at 500 μg/mL 72 h post-transfection, and stable pools were obtained after 3 weeks of selection. To measure NHEJ efficiency in stable pools, cells (IGROV-1, SK-OV-3) were first pre-incubated with different concentrations of Panobinostat or NU-7026 for 24 h. Then, 10^6^ cells were co-transfected with 5 µg of a plasmid encoding the I-SceI endonuclease and 0.5 µg of pDsRed-N1 to correct for differences in transfection efficiencies. After transfection, cells were incubated again with the same concentration of Panobinostat or NU-7026 for 72 h. Transfections were performed as described before. Green (EGFP) and Red (DsRed) fluorescences were measured by flow cytometry, and the NHEJ efficiency was calculated as the ratio of EGFP+ to DsRed+ cells and then normalized to the untreated control.

### 4.6. Cell Proliferation Assay

OCCLs were seeded into 96-well plates (4 × 10^3^ cells/well), and after 24 h of culture, they were incubated in the absence or the presence of different concentrations of KU-57788, NU-7026, or SCR7 pyrazine for 24, 48, or 72 h. Cell viability was determined using 3-(4,5-dimethylthiazol-2-yl)-2,5-diphenyl-2H-tetrazolium bromide (MTT) (Sigma-Aldrich). MTT was dissolved in PBS (5 mg/mL), and 10 µL of the solution was added to each well. After 1 h of incubation, the medium was aspirated, and formazan crystals were dissolved in DMSO (100 µL/well). Absorbance was measured at 570 nM in a plate reader (Ultra Evolution, Tecan, Männedorf, Switzerland). The half-maximal inhibitory concentration (IC50) was calculated using GraphPad Prism 9.

### 4.7. Cell Cycle Analysis

OCCLs were seeded into 12-well plates (7.5 × 10^4^ cells/well), and after 24 h of culture, they were treated with KU-57788, NU-7026, or SCR7 pyrazine for 24, 48, and 72 h. Then, they were fixed in 70% ethanol and stored at 4 °C for later use. Cells were then rehydrated with PBS, stained with 50 µg/mL of propidium iodide (PI) (Sigma-Aldrich), and treated overnight with 100 µg/mL of RNase A in the dark (Sigma-Aldrich). Cell cycle profiles were obtained by flow cytometry using a BD Accuri™ C6 Plus Flow Cytometer (BD Biosciences, San Jose, CA, USA). Data were analyzed with BD Accuri™ C6 Software (version 1.0.264.21).

### 4.8. Apoptosis Assay

OCCLs were seeded into 12-well plates (4.5 × 10^4^ cells/well), and after 24 h of culture, they were treated with chloroquine, Panobinostat, N-Acetyl-L-Cysteine, and/or NHEJi for 72 h, and then apoptosis was measured using annexin V-fluorescein isothiocyanate/propidium iodide (PI) double-staining (Immunostep, Salamanca, Spain) according to the manufacturer’s procedure. The numbers of apoptotic cells were determined by flow cytometry. The synergism of the combination was quantified using Compusyn Software (ComboSyn, Inc., Paramus, NJ, USA), which is based on the Chou–Talalay method [87] and calculates a combination index (CI) with the following interpretation: CI > 1: antagonistic effect; CI = 1: additive effect; CI < 1 synergistic effect.

### 4.9. Immunofluorescence

OCCLs were plated on round glass coverslips (12 mm diameter) (150,000 cells/well in 6-well plates), and, after 24 h of culture, cells were submitted to different treatments, fixed with 4% paraformaldehyde for 10 min, and permeabilized using 0.5% Triton X-100 (Boehringer Mannheim, Mannheim, Germany) in PBS for 10 min. Then, samples were blocked in 10% BSA in PBS for 30 min and incubated with phospho-H2AX antibody (1:1000, 05-636, Sigma-Aldrich) for 90 min. After washing, coverslips were incubated with fluorescent secondary antibodies (1:400, Alexa Fluor 488 goat anti-mouse IgG, Molecular Probes, Invitrogen) for 1 h. DAPI (dihydrochloride of 4′,6-diamidino-2-phenylindole, Roche) was used to visualize the nuclei. Mowiol reagent (Calbiochem, San Diego, CA, USA) was used to fix preparations on slides. Cells were then analyzed by confocal microscopy (63×) using a LEICA SP5 microscope DMI-6000V model coupled to a Leica Application Suite X software computer (version 3.5.7.23225).

### 4.10. Statistical Analysis

Differences between the results obtained from treated and nontreated cells were assessed for statistical significance using Student’s unpaired two-tailed *t*-test with jamovi for Mac version 2.2.5 [88]. ANOVA with Tukey’s post hoc test was used when more than two groups were compared. Data are presented as mean ± standard deviations. Statistical significance was concluded for values of *p* ≤ 0.05 (*** *p* < 0.001, ** *p* < 0.01, * *p* < 0.05).

## 5. Conclusions

Altogether, our study indicates that the NHEJ pathway plays an essential role in repairing CQ-induced DNA double-strand breaks, which explains the synergy found between CQ and NHEJ inhibitors. Moreover, the inhibition of both DNA repair pathways, NHEJ and HR, highly increased the synergistic effect. Together with our previous report [30], these findings suggest that the combination of CQ and HR inhibitors or NHEJ inhibitors should be explored as a new strategy for the treatment of OC.

## Figures and Tables

**Figure 1 ijms-23-07518-f001:**
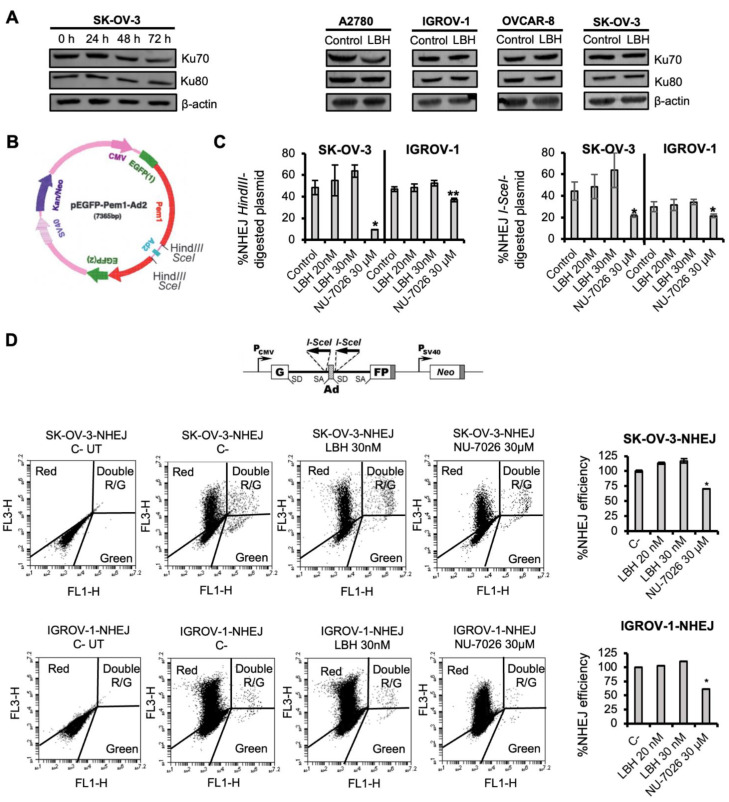
Panobinostat does not affect DSB repair by NHEJ. (**A**) Left panel: time-response of NHEJ-related proteins (Ku70, Ku80) after LBH treatment (50 nM) in SK-OV-3 cell line. Right panel: Ku70 and Ku80 expression after 24 h of LBH treatment (20 nM) in OCCLs. β-actin was used as a loading control. (**B**) Map of pEGFP-Pem1-Ad2. An Ad2 exon is present in the middle of the Pem1 intron, and efficient splicing inactivates the GFP activity and makes the starting substrate GFP-negative. However, both sides of the Ad2 exon present *HindII/I-SceI* restriction sites. Cleavage with either of these endonucleases removes the Ad exon, and upon successful intracellular plasmid circularization, GFP expression is restored and can be quantified by flow cytometry [47]. (**C**) Percentage of NHEJ using *HindIII*- or *I-SceI*-digested plasmid in IGROV-1 and SK-OV-3 cell lines. Cells were pre-treated or not with the indicated doses of LBH or NU-7026, transfected with the linearized pEGFP-Pem1-Ad2 or supercoiled pEGFP-Pem1 together with the pDSRed plasmid, and incubated again with LBH or NU-7026 for 72 h. The percentage of NHEJ was calculated as described in the Materials and Methods. (**D**) Top panel: Map of NHEJ-C reporter construct [48]. Bottom panel: Dot plots of nontransfected SK-OV-3 and IGROV-1 cells carrying the NHEJ reporter cassette, and the same cell lines co-transfected with 5 µg of an *I-SceI* endonuclease-expressing plasmid and 0.5 µg of pDsRed2-N1. The latter were incubated in the presence or absence (C-) of LBH or NU-7026 for an additional 72 h. Correct NHEJ repair restored the GFP gene, which was detected as GFP+ cells. NHEJ efficiency was calculated as the ratio of GFP+ to DsRed+ cells and then normalized to the untreated control. C-: negative control (untreated cells). Data are the mean of three independent experiments. Error bars represent the SD (** *p* < 0.01, * *p* < 0.05 compared to controls).

**Figure 2 ijms-23-07518-f002:**
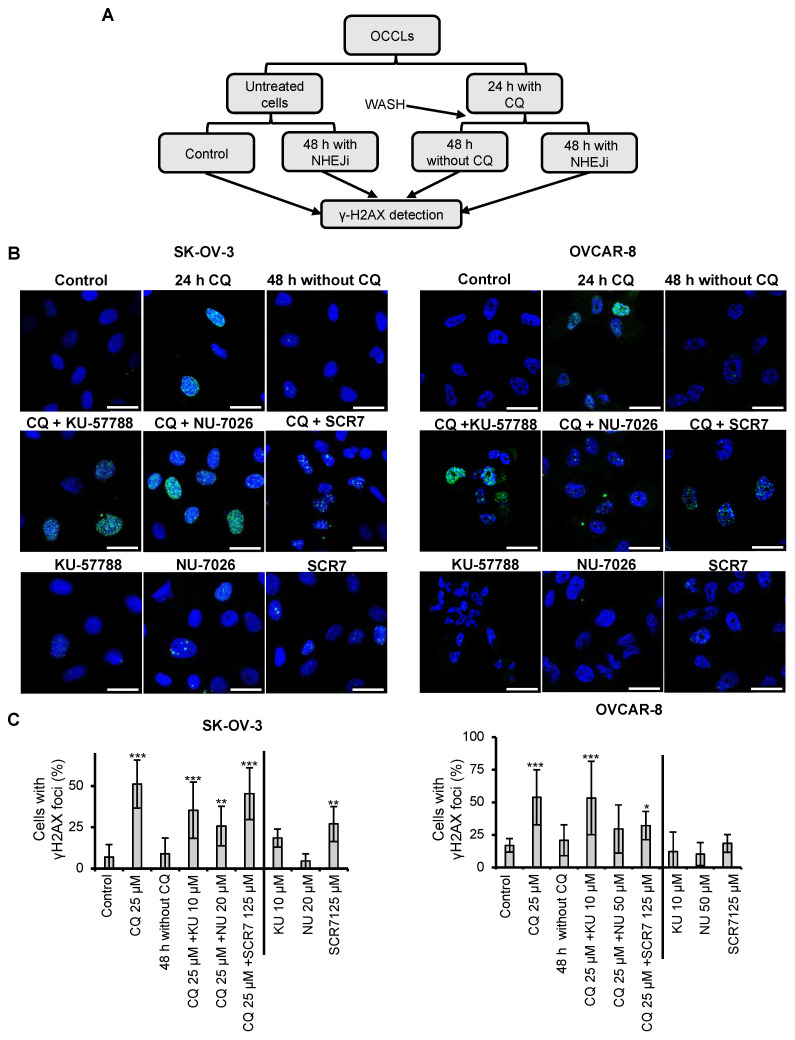
Effect of NHEJ inhibitors in the repair of DSBs generated by chloroquine. (**A**) Scheme of the experiment. OCCLs were treated for 24 h with 25 µM of CQ. Then, CQ was removed by washing, and cells were treated or not with NHEJi for 48 h. Part of the cells were left untreated or treated only with NHEJi. (**B**) Representative immunofluorescence microscopy images of OCCLs treated with the indicated conditions and stained with DAPI (blue) and anti-γH2AX (green) to visualize DNA DSBs. Scale bar: 25 µM. (**C**) Percentage of cells exhibiting γH2AX foci (>5 foci/cell) in the presence of CQ, NHEJi, or both. A minimum of 50 cells per condition were analyzed in three independent experiments. Error bars represent the SD (*** *p* < 0.001; ** *p* < 0.01; * *p* < 0.05, compared to CQ-treated cells 48 h after CQ removal).

**Figure 3 ijms-23-07518-f003:**
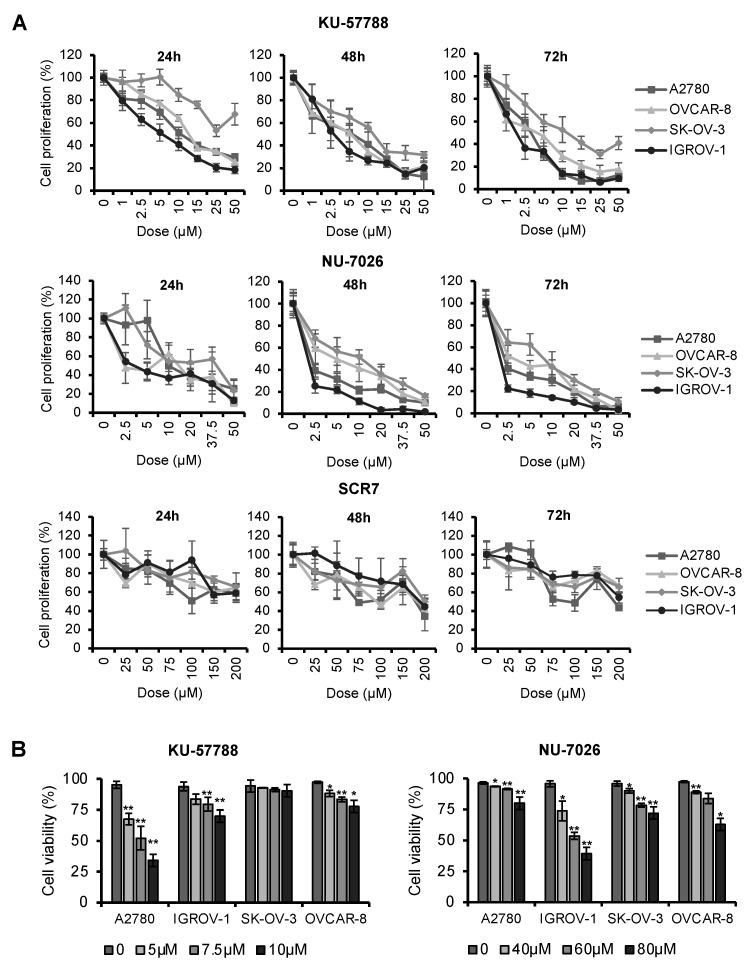
Effect of NHEJi in proliferation and cell viability. (**A**) Cell proliferation after treatment with the indicated doses of KU-57788, NU-7026, or SCR7 pyrazine for 24, 48, and 72 h. Data are the mean of three independent experiments. (**B**) Cell viability after the treatment with the indicated doses of KU-57788 and NU-7026. The percentage of live cells was determined by Annexin V/PI staining. Data are the mean of at least three independent experiments. Error bars represent the SD. Differences between the results obtained from treated and nontreated cells were assessed for statistical significance using Student’s unpaired two-tailed *t*-test (** *p* < 0.01 and * *p* < 0.05 compared to untreated cells).

**Figure 4 ijms-23-07518-f004:**
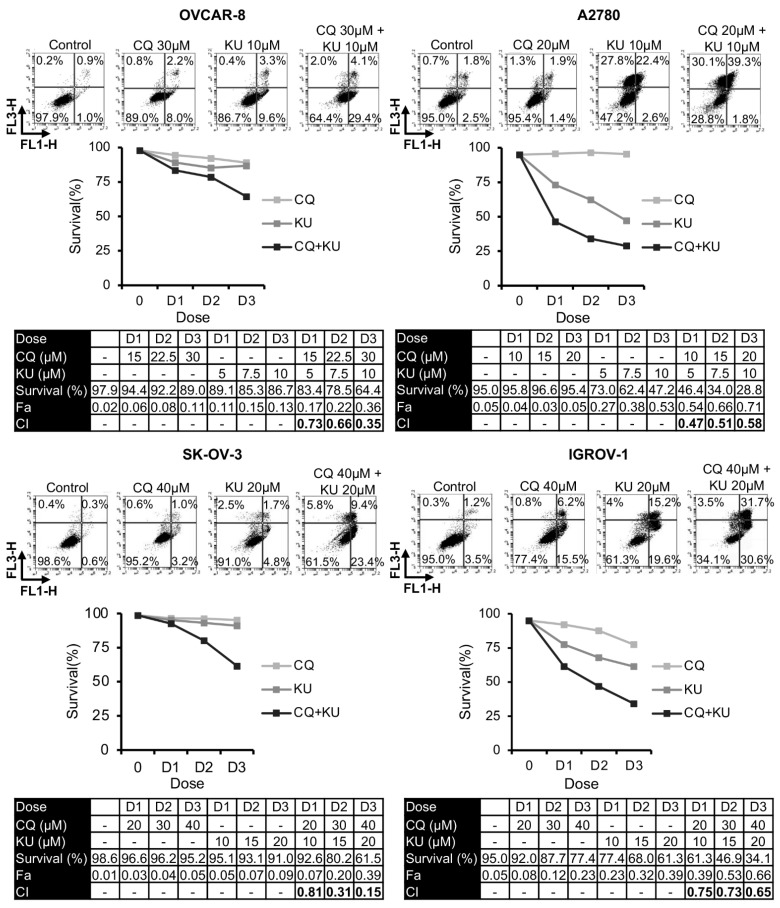
Synergistic effect of chloroquine and KU-57788 in OCCLs. Cells were exposed for 72 h to the indicated concentrations of KU and CQ at a constant ratio, and the percentage of apoptotic cells was assessed by flow cytometry (after cell staining with annexin V and propidium iodide). CI values, calculated using Compusyn Software (version 1.0), are shown.

**Figure 5 ijms-23-07518-f005:**
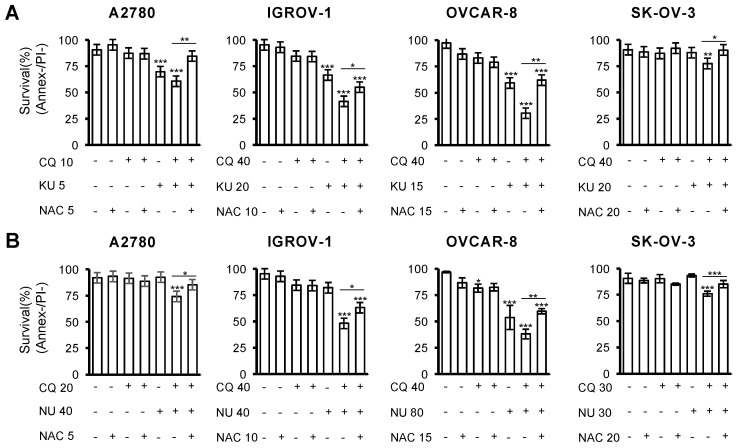
CQ/NHEJi lethality depends on ROS production. (**A**) Cells were exposed for 72 h to the indicated concentrations of CQ (µM) and KU-57788 (µM), or (**B**) NU-7026 (µM) and ROS scavenger NAC (mM), and the percentage of apoptotic cells was measured after cell staining with annexin V and propidium iodide by flow cytometry. Data are the mean of at least three independent experiments. Error bars represent the SD (*** *p* < 0.001, ** *p* < 0.01, and * *p* < 0.05, compared to C-, unless otherwise specified; ANOVA with Tukey’s post hoc test was used for the analysis of these results).

**Figure 6 ijms-23-07518-f006:**
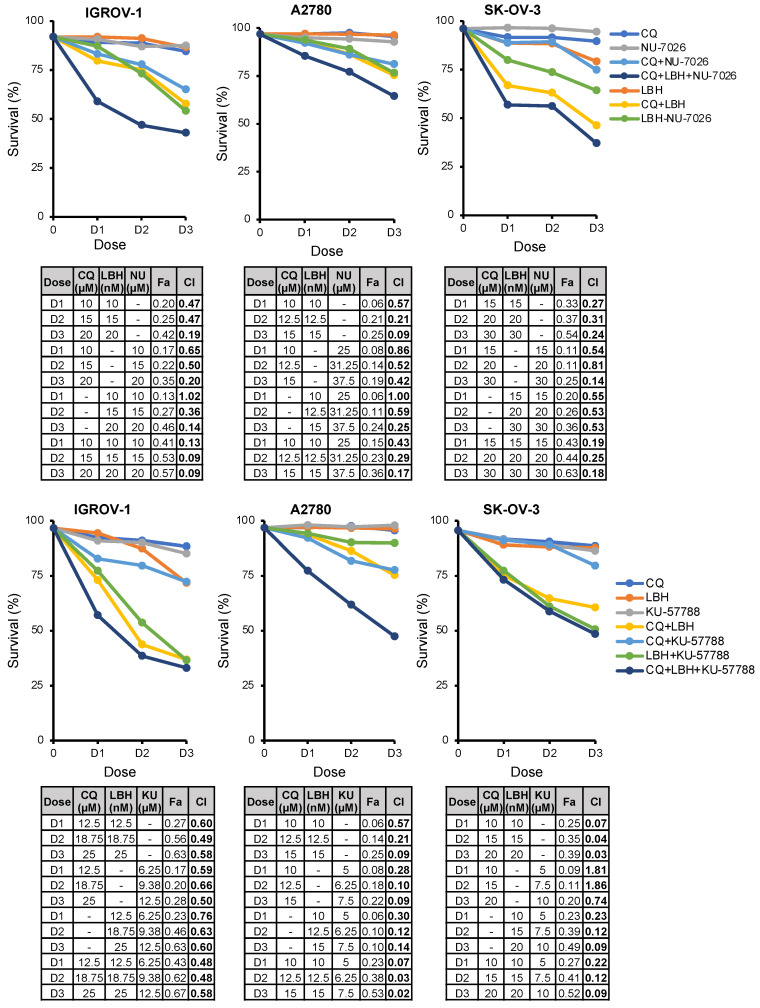
The triple-combination CQ-NHEJi-LBH exerts a more potent synergy than double combinations. Cells were exposed for 72 h to the indicated concentrations of LBH, CQ, and KU-57788/NU-7026 at a constant ratio, and the percentage of apoptotic cells was assessed by flow cytometry (after cell staining with annexin V and propidium iodide). CI values, calculated using Compusyn Software, are shown.

**Table 1 ijms-23-07518-t001:** IC50 values for NHEJi. Best-fit values for IC50 values and interval in which IC50 is included. These values were calculated using GraphPad Prism software (version 9).

Cell Line	A2780	IGROV-1	OVCAR-8	SK-OV-3
Inhibitors	IC50 (95% CI)	IC50 (95% CI)	IC50 (95% CI)	IC50 (95% CI)
KU-57788	2.56 µM (2.25–2.89 µM)	1.86 µM (1.62–2.12 µM)	3.42 µM (2.81–4.14 µM)	11.07 µM (9.07–13.51 µM)
NU-7026	2.51 µM (2.10–2.97 µM)	0.96 µM (0.81–1.12 µM)	4.27 µM (3.45–5.26 µM)	7.30 µM (6.38–8.36 µM)
SCR7	204.5 µM (129.7–348.8 µM)	329.8 µM (253.1–446.9 µM)	305.4 µM (223.2–439.4 µM)	269.5 µM (202.3–373.1 µM)

CI: Confidence interval.

## Data Availability

Not applicable.

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
