# Peer review of "Chloroquine-Induced DNA Damage Synergizes with Nonhomologous End Joining Inhibition to Cause Ovarian Cancer Cell Cytotoxicity"

_ijms, 2022, doi:10.3390/ijms23147518_

Round 1

Reviewer 1 Report

The authors have presented a research work towards establishing a potential combination treatment strategy for ovarian cancer. Authors have established the role of NHEJ inhibition post CQ-stimulated DNA damage and highlighted the role of this mechanism in lesion repair. The synergistic effect of the combinatory approach was also studied with respect to apoptosis induction. Authors have also tested the effects of Panobinostat in the combination strategy and demonstrated the synergistic potential of CQ-NHEJi-LBH. Overall, the research context looks promising and the study indeed can be considered as a preliminary study however in vivo data needs to be taken into account for establishing the utility of the approach. There are few concerns with regard to the methodology and data validation, and questions are added below:

1.       Line 38-39, introduction: ‘after further treatments with different chemotherapeutic agents- please specify the chemo drugs here as this is a relevant statement and provide more references to support the statement.

2.       Has Chloroquine been approved yet as an ‘anti-cancer drug ‘? It has been approved for malaria treatment and also to treat certain auto immune conditions. Hence, authors could provide few more insights in this direction in the introductory section.

3.       Was there any specific reason for choosing Panobinostat in the context of ovarian cancer treatment with respect to its mechanism of action and ovarian cancer metastasis? Please provide literature support.

4.       It would be really helpful if authors could present a detailed schematic to explain the potential mechanism involved in the proposed combinatory approach as a separate figure – this would enhance the readability of the manuscript as there are multiple aspects of the mechanism which are being investigated in detail.

5.       Please explain the rationale behind using four different ovarian cancer cell lines for the study as they differ in different characteristics. Authors have indicated this with regard to histopathological attributes in discussion section but it needs to be separately indicated in the introductory section.

6.       How was the data reproducibility verified for the in vitro assays such as NHEJ assays, cell proliferation assay, cell cycle analysis, apoptosis assay and immunofluorescence? Please comment on details such as number of wells seeded, experimental and technical replicates as these need to be taken into account for validating the data.

7.       The data presented in Figure 2B need to have scale bars and better resolution to better interpret the results. Figure 2 legend indicates that mean was based on atleast 50 cells/ cell line and condition. This is unclear and needs to be better defined.

8.       How was %cells exhibiting γH2AX foci calculated? (Figure 2C)

9.       How was the statistical significance confirmed for Figure 3B data? Please explain the test used with all factors taken for consideration. This also needs to be indicated in the figure legend – including details such as test used, n, p value.

10.   What could be the reason for KU-57788 being required in lower doses compared to NU-7026 to induce apoptosis? (Line 179, page 6)

11.   “The triple combination CQ-LBH-NHEJi exerted a strong synergistic effect that was higher than double combinations, especially when NU-7026 was employed, with the exception of SK-OV-3 cell line treated with CQ-LBH-KU-57788”- lines 210-213 in page 7, interesting observation for SKOV3 and hence would suggest authors to give a possible explanation.

12.   Please explain the statistical method used for Figure 5 data analysis. Mention this in the legend as well.

13.   It is really good that authors have indicated the limitations of the current study and the need for in vivo studies in discussion section. Could the solubility issue be explained in a better way (line 312, page 12)?

14.   It would be really great if authors can briefly point out that  the observations from the current study have been based on ‘in vitro experimental approach and efficacy’  in abstract as well as in the introductory section. 

Author Response

The authors have presented a research work towards establishing a potential combination treatment strategy for ovarian cancer. Authors have established the role of NHEJ inhibition post CQ-stimulated DNA damage and highlighted the role of this mechanism in lesion repair. The synergistic effect of the combinatory approach was also studied with respect to apoptosis induction. Authors have also tested the effects of Panobinostat in the combination strategy and demonstrated the synergistic potential of CQ-NHEJi-LBH. Overall, the research context looks promising and the study indeed can be considered as a preliminary study however in vivo data needs to be taken into account for establishing the utility of the approach. There are few concerns with regard to the methodology and data validation, and questions are added below:

  1. Line 38-39, introduction: ‘after further treatments with different chemotherapeutic agents- please specify the chemo drugs here as this is a relevant statement and provide more references to support the statement.

Following the reviewer's recommendations we have specified the chemo drugs and added more references [line 39-40, references 5-7].

  1. Has Chloroquine been approved yet as an ‘anti-cancer drug ‘? It has been approved for malaria treatment and also to treat certain auto immune conditions. Hence, authors could provide few more insights in this direction in the introductory section.

As far as we know, CQ has not yet been approved as anti-cancer drug. However, it is considered a potential anticancer agent. We have modified the information shown in the introduction as follows: “In addition, CQ is considered a potential anticancer agent; it has been described to reduce hypoxia, cancer cell invasion, and metastasis [10]. Moreover, several preclinical results and clinical trials have shown that this compound, in combination with other antitumor drugs, increases the cytotoxic effect and sensitizes tumor cells to radiotherapy or chemotherapy [9, 11-23]”, (lines 46 to 50).

  1. Was there any specific reason for choosing Panobinostat in the context of ovarian cancer treatment with respect to its mechanism of action and ovarian cancer metastasis? Please provide literature support.

We have chosen Panobinostat for several reasons: First, because it is structurally similar to vorinostat but exhibits a higher potency, ten times superior to vorinostat (doi: 10.2147/OTT.S30773). Second, because Panobinostat has proven their efficacy in vitro and in vivo in ovarian cancer [doi: 10.1371/journal.pone.0158208), which might be related to the fact that most epithelial ovarian carcinomas show epigenetic changes and overexpression of histone deacetylases (HDACs) (doi:10.1016/j.gore.2017.03.007). And third, because it decreases HR repair and approximately half of all women diagnosed with ovarian cancer are HR-proficient. In this regard, a recent study indicates that Panobinostat in combination with Olaparib reduced peritoneal metastases and tumor burden in a syngeneic mouse model of ovarian cancer (doi: 10.1016/j.neo.2021.12.002).

  1. It would be really helpful if authors could present a detailed schematic to explain the potential mechanism involved in the proposed combinatory approach as a separate figure – this would enhance the readability of the manuscript as there are multiple aspects of the mechanism which are being investigated in detail.

We thank the reviewer for this suggestion. We have prepared a graphical abstract that explain the proposed mechanisms.

  1. Please explain the rationale behind using four different ovarian cancer cell lines for the study as they differ in different characteristics. Authors have indicated this with regard to histopathological attributes in discussion section but it needs to be separately indicated in the introductory section.

The information requested has also been indicated in the introductory section (line 78).

  1. How was the data reproducibility verified for the in vitro assays such as NHEJ assays, cell proliferation assay, cell cycle analysis, apoptosis assay and immunofluorescence? Please comment on details such as number of wells seeded, experimental and technical replicates as these need to be taken into account for validating the data.

Better explanations of the statistics and number of replicates have been added in materials and methods and in the correspondent figure legends. Thanks

  1. The data presented in Figure 2B need to have scale bars and better resolution to better interpret the results. Figure 2 legend indicates that mean was based on atleast 50 cells/ cell line and condition. This is unclear and needs to be better defined.

The reviewer is right, the scale bars have been modified and have now a better resolution. Statistics have been incorporated to Figure 2C and also the following sentence: “A minimum of 50 cells per condition were analyzed in three independent experiments. Error bars represent the SD (***p<0.001; **p<0.01; p<0.05, compared to CQ-treated cells 48 h after CQ removal)” (lines 163-165).

  1. How was %cells exhibiting γH2AX foci calculated? (Figure 2C)

The percentage of cells with γH2AX foci (> 5 foci/cell) were calculated by counting the number of γH2AX foci per cell.  A minimum of 50 cells per experiment and cell line were counted in three independent experiments.

  1. How was the statistical significance confirmed for Figure 3B data? Please explain the test used with all factors taken for consideration. This also needs to be indicated in the figure legend – including details such as test used, n, p value.

A better explanation of the statistics, number of replicates and test used have been incorporated (lines 199-204).

  1. What could be the reason for KU-57788 being required in lower doses compared to UN-7026 to induce apoptosis? (Line 179, page 6)

This difference could be explained by the higher potency of KU-57788 to inhibit DNA-PK activity(https://www.medchemexpress.com/KU-57788.html;https://www.medchemexpress.com/NU-7026.html)   

  1. “The triple combination CQ-LBH-NHEJi exerted a strong synergistic effect that was higher than double combinations, especially when UN-7026 was employed, with the exception of SK-OV-3 cell line treated with CQ-LBH-KU-57788”- lines 210-213 in page 7, interesting observation for SKOV3 and hence would suggest authors to give a possible explanation.

We agree with the reviewer in that it is an interesting observation. A possible explanation has been added [lines 313-324]: “We found that inhibition of both DNA repair pathways, NHEJ and HR, highly increased the synergistic effect with CQ, except in the SK-OV-3 cell line. This cell line bears a TP53 homozygous mutation (Expasy, Cellosarous), whereas the A2780 cell line is wild-type for TP53 and the IGROV-1 cell line has a heterozygous mutation (c.377A>G, Expasy, Cel-losarous) classified as an uncertain significance variant (ClinVar= VCV000458541). Therefore, p53 loss in SK-OV-3 might avoid apoptosis induction even in the absence of both DNA repair pathways. In this regard, prior studies have demonstrated that p53-deficient cells have lost the balance between HR activation and repression[65] and also that DNA-PK inhibition leads to the use of an alternative end-joining mechanism to promote cell viability in response to DSBs[66]. Abrogation of both pathways might po-tentiate alternative mechanisms promoting cell survival; however, testing this hypothesis would require further research”.

  1. Please explain the statistical method used for Figure 5 data analysis. Mention this in the legend as well.

The statistical method has been added to Figure 5 legend (line 248).

  1. It is really good that authors have indicated the limitations of the current study and the need for in vivo studies in discussion section. Could the solubility issue be explained in a better way (line 312, page 12)?

The solubility issue has been better explained in lines 342 to 346 ("The advantages of using one of these NHEJi versus others that have been developed, in combination with CQ, need to be evaluated in in vivo experiments. In this regard, although KU-57788 has been shown to provide excellent proof of principle in vitro and in vivo chemo- and radio-sensitization, it is known to present a limited aqueous solubility which restricts its further development [71]”).

  1. It would be really great if authors can briefly point out that the observations from the current study have been based on ‘in vitro experimental approach and efficacy’  in abstract as well as in the introductory section.

The reviewer is right, we have added in the abstract and in the introductory section that our observations correspond to in vitro studies (lines 21 and 27, Abstract; Line 80 Introduction).

Reviewer 2 Report

Ovejero-Sánchez et al. present evidence that chloroquine (CQ) and NHEJ inhibitor synergize to cause OC cytotoxicity, which is dependent on ROS. This is an extension of their previous work showing a synergy between CQ and HR inhibitor LBH in OC cytotoxicity. They also show that CQ-NHEJi-LBH is more potent than CQ-NHEJi and CQ-LBH. The findings are consistent with the notion that CQ may cause cytotoxicity via making ROS mediated dsDNA breaks and are not surprising. However, they are clearly clinically relevant regarding improvement of OC therapy.  

If possible, the authors may want to discuss different sensitivities of OC lines to distinct treatments as they find.

Fig. 2B             annotation is not self-explanatory

Fig. 3A             organization is good for comparing different OV lines, but not   for comparing growth of a line as a function of time  

Minor points

Line 71            lacks a period after “protein”

89                    Fig. 1B should be 1A

90                    “ex-tra” should be “extra-“

91                    “abil-ity” should be “ability”

131                  “this last” should be “ the latter”

317-318           o-f 

Author Response

Ovejero-Sánchez et al. Present evidence that chloroquine (CQ) and NHEJ inhibitor synergize to cause OC cytotoxicity, which is dependent on ROS. This is an extension of their previous work showing a synergy between CQ and HR inhibitor LBH in OC cytotoxicity. They also show that CQ-NHEJi-LBH is more potent than CQ-NHEJi and CQ-LBH. The findings are consistent with the notion that CQ may cause cytotoxicity via making ROS mediated dsDNA breaks and are not surprising. However, they are clearly clinically relevant regarding improvement of OC therapy. 

If possible, the authors may want to discuss different sensitivities of OC lines to distinct treatments as they find.

Differences between cell lines and their response to different treatments have been discussed on the manuscript. In relation to differences in cell cycle distribution between OCCLs the following information has been added to results: Cytotoxicity to NU-7026 in OVCAR-8 and SK-OV-3 was lower than the observed for A2780 and IGROV-1, which is consistent with the absence of G1 arrest in those cell lines. (Lines 189-191, Results). Differences in the sensitivity to triple combinations have been also discussed (lines 313-324, Discussion): “We found that inhibition of both DNA repair pathways, NHEJ and HR, highly increased the synergistic effect with CQ, except in the SK-OV-3 cell line. This cell line bears a TP53 homozygous mutation (Expasy, Cellosarous), whereas the A2780 cell line is wild-type for TP53and the IGROV-1 cell line has a heterozygous mutation (c.377A>G, Expasy, Cel-losarous) classified as an uncertain significance variant (ClinVar= VCV000458541). Therefore, p53 loss in SK-OV-3 might avoid apoptosis induction even in the absence of both DNA repair pathways. In this regard, prior studies have demonstrated that p53-deficient cells have lost the balance between HR activation and repression[65] and also that DNA-PK inhibition leads to the use of an alternative end-joining mechanism to promote cell viability in response to DSBs[66]. Abrogation of both pathways might po-tentiate alternative mechanisms promoting cell survival; however, testing this hypothesis would require further research”.

Fig. 2B             annotation is not self-explanatory

We have added Figure 2B and Figure 2C when they were necessary to improve understanding.

Fig. 3ª             organization is good for comparing different OV lines, but not   for comparing growth of a line as a function of time 

Following the recommendation of the reviewer we have created a new supplementary figure (Figure S1) that includes the graphics with the growth of each OC cell line after 24, 48 and 72 hours of culture.

Minor points

Line 71            lacks a period after “protein”

89                    Fig. 1B should be 1ª

90                    “extra” should be “extra-“

91                    “abil-ity” should be “ability”

131                  “this last” should be “ the latter”

317-318           o-f

We thank the reviewer; the manuscript has been revised and several typos have been corrected.

Reviewer 3 Report

Because ovarian cancer causes high mortality, it is necessary to develop effective ways to cure this type of cancer. Dr Herrero and her colleagues showed that a combination of chloroquine and DNA-PK and HDAC inhibitors (CQ-NHEJi-Panobinostat) synergistically exhibit toxic effects on cancer cells. The data are convincing, and the paper is well written. However, adding the normal cell control will increase the impact of this paper.

Major points:

1.       It is expected to see that the inhibition of NHEJ and HR pathways is toxic to cells containing DNA double-strand breaks. Therefore, it is important to show whether the CQ-NHEJi-Panobinostat treatment is more effective on ovarian cancer than normal or other cancer cells.

Minor points:

1.       Line 67. ‘On the other hand,’. Delete.

2.       Line 71. ‘proteinWe’ must be ‘protein. We’.

3.       Line 90 and 91. ‘ex-trachromosomal’ and ‘abil-ity’ must be ‘extrachromosomal’ and ‘ability’.

4.       Line 100-101. ‘HDACi does not alter DSB repair by NHEJ (Figure 1C)’. It is essential to mention that LBH at these concentrations was effective in other experiments (Figure 6).

5.       Figure 1B. Can the authors briefly explain what Pem1 is in the plasmid in the figure legend?

6.       Figure 2 legend title. ‘Panobinostat does not affect DSB repair by NHEJ.’ is not relevant.

7.       Figure 2B. Are anti-γH2AX signals in green?

8.       Figure 2C. ‘the mean of the analysis of at least 50 cells’ is unclear. Did the authors do multiple sets of the analysis? 

9.       Figure 2C. Statistical evaluation is required.

10.    Line 161. Can the authors discuss why KU-57788 and NU-7026 were more potent than SCR7 pyrazine?

11.    Line 168-170 and 210-213. Can the authors discuss why the different cell lines showed different phenotypes?

12.    Line 262-267 and 305-309. The long sentences were difficult to understand.

Author Response

Because ovarian cancer causes high mortality, it is necessary to develop effective ways to cure this type of cancer. Dr Herrero and her colleagues showed that a combination of chloroquine and DNA-PK and HDAC inhibitors (CQ-NHEJi-Panobinostat) synergistically exhibit toxic effects on cancer cells. The data are convincing, and the paper is well written. However, adding the normal cell control will increase the impact of this paper.

Major points:

  1. It is expected to see that the inhibition of NHEJ and HR pathways is toxic to cells containing DNA double-strand breaks. Therefore, it is important to show whether the CQ-NHEJi-Panobinostat treatment is more effective on ovarian cancer than normal or other cancer cells.

We thank the reviewer for this suggestion. We are actually evaluating and comparing the effects of double and triple combinations in differing cell lines derived from other solid tumors and are also obtaining promising results. We plan to write a manuscript with all the findings. Regarding normal cells and following the reviewer´s recommendation we have analyzed the effect of the drugs under study in the human embryonic kidney HEK293T cells. We also found synergistic effects in both, double and triple combinations, which is not surprising since they have a very high proliferation rate, even higher than OC cell lines. It is known that cells with high rates of proliferation are more susceptible to DNA damage (doi: 10.1016/j.molonc.2011.07.006) and exhibit higher levels of reactive oxygen species. This last can result from increased metabolic activity since ATP synthesis produces ROS during normal oxygen metabolism (doi:10.1007/s11010-019-03667-9). Several works have combined genotoxic agents with DNA repair inhibitors in vivo and have found cytotoxic effects in tumor cells and lower toxicity in normal cells. Therefore, CQ in combination with DNA repair inhibitors could drive DNA damage and oxidative pressure to toxic levels selectively or preferentially in tumor cells. The advantages of using CQ together with NHEJ inhibitors and/or HR inhibitors need to be evaluated in in vivo experiments. Depending on the results, encapsulation or targeted delivery of the drugs could also be explored to recapitulate the efficacy of the combinations shown in this work.

Minor points:

  1. Line 67. ‘On the other hand,’. Delete.

Corrected

  1. Line 71. ‘proteinWe’ must be ‘protein. We’.

Corrected

  1. Line 90 and 91. ‘ex-trachromosomal’ and ‘abil-ity’ must be ‘extrachromosomal’ and ‘ability’.

We thank the reviewer; we have revised the manuscript and correct several typos.

  1. Line 100-101. ‘HDACi does not alter DSB repair by NHEJ (Figure 1C)’. It is essential to mention that LBH at these concentrations was effective in other experiments (Figure 6).

It is true, this consideration was only indicated for the assays using NHEJ-reporter cell lines. The following sentence has been added to the results: “LBH-doses used in these assays were effective in SK-OV-3 and IGROV-1 cell lines causing a 50% reduction of cell survival”. We have included this indication for extrachromosomal assays (lines 106-107).

  1. Figure 1B. Can the authors briefly explain what Pem1 is in the plasmid in the figure legend?

We have explained what Pem1 and Ad2 are in the figure legend as requested.

  1. Figure 2 legend title. ‘Panobinostat does not affect DSB repair by NHEJ.’ is not relevant.

The reviewer is right. This tittle has been deleted from Figure 2 legend. Thanks.

  1. Figure 2B. Are anti-γH2AX signals in green?

Yes, we have included this information in Figure 2 legend.

  1. Figure 2C. ‘the mean of the analysis of at least 50 cells’ is unclear. Did the authors do multiple sets of the analysis?

The percentage of cells with γH2AX foci (> 5 foci/cell) were calculated by counting the number of γH2AX foci per cell.  A minimum of 50 cells per experiment and cell line were counted in three independent experiments and the mean of these experiments was represented.

  1. Figure 2C. Statistical evaluation is required.

Statistics have been incorporated to figure 2C and also the following sentence: “A minimum of 50 cells per condition were analyzed in three independent experiments. Error bars represent the SD (***p<0.001; **p<0.01; p<0.05, compared to CQ-treated cells 48 h after CQ removal).

  1. Line 161. Can the authors discuss why KU-57788 and NU-7026 were more potent than SCR7 pyrazine?

KU-57788 and NU-7026 inhibit DNA-PK activity whereas SCR7 pyrazine inhibits DNA ligase IV. A higher doses of DNA ligase IV inhibitor might be necessary to obtain the same effect than with DNA-PKs inhibitors. In fact, IC50s reported by the supplier are higher than those reported by the DNA-PK inhibitors. It has also been described that OC cells required higher doses of SCR7 than other cancer cell lines [75].

(https://www.medchemexpress.com/KU-57788.html;https://www.medchemexpress.com/NU-7026.html; https://www.medchemexpress.com/scr7-pyrazine.html)  

  1. Line 168-170 and 210-213. Can the authors discuss why the different cell lines showed different phenotypes?

In relation to differences in cell cycle distribution between OCCLs the following information has been added to results: Cytotoxicity to NU-7026 in OVCAR-8 and SK-OV-3 was lower than the observed for A2780 and IGROV-1, which is consistent with the absence of G1 arrest in those cell lines (lines 189-191, Results). Differences in the sensitivity to triple combinations have been also discussed (lines 313-324, Discussion): “We found that inhibition of both DNA repair pathways, NHEJ and HR, highly increased the synergistic effect with CQ, except in the SK-OV-3 cell line. This cell line bears a TP53 homozygous mutation (Expasy, Cellosarous), whereas the A2780 cell line is wild-type for TP53and the IGROV-1 cell line has a heterozygous mutation (c.377A>G, Expasy, Cel-losarous) classified as an uncertain significance variant (ClinVar= VCV000458541). Therefore, p53 loss in SK-OV-3 might avoid apoptosis induction even in the absence of both DNA repair pathways. In this regard, prior studies have demonstrated that p53-deficient cells have lost the balance between HR activation and repression[65] and also that DNA-PK inhibition leads to the use of an alternative end-joining mechanism to promote cell viability in response to DSBs[66]. Abrogation of both pathways might po-tentiate alternative mechanisms promoting cell survival; however, testing this hypothesis would require further research”.

  1. Line 262-267 and 305-309. The long sentences were difficult to understand.

We thank the reviewer for the suggestion; these sentences have been changed for a better understanding.

Reviewer 4 Report

The study entitled “Chloroquine-induced DNA damage synergizes with non-homologous end joining inhibition to cause ovarian cancer cell cytotoxicity”

investigated the potential synergistic effects of CQ + NHEJi and CQ + NHEJi + LBH. Manuscript attempted to build on the previous findings of the group and others. Authors need to address several major and minor concerns outlined below.

Major points: 

·      Authors must include colony forming assay for evaluation of cell survival. 

·      Non-cancer cell line must be included as an important control for all the assays. Author mentioned that “….in agreement with our results, in vivo xenografts experiments using this cell line revealed that treatment with SCR7 resulted in regression of tumors with no obvious adverse effects” but references and/or data to support this claim are not provided. 

·      Figure 2: For panel C, authors must include error bars, statistics and number of biological replicates. Based on the description provided in the figure legend, it seems that data presented are from only one experiment, which is not acceptable. 

·      Figure 3: For panel C, authors must provide statistics and number of biological replicates. 

·      Figure 4: Authors must include error bars, statistics and number of biological replicates. 

·      Figure 6: Authors must include error bars, statistics and number of biological replicates

Minor point: 

·      There are several typos throughout the manuscript, authors should correct all of them. Few are listed here;

Page 2, line 71, …. proteinWe

Page 2, line 90, .. ex-trachromosomal

Page 2, line 91,… measuring cell abil-ity

Author Response

The study entitled “Chloroquine-induced DNA damage synergizes with non-homologous end joining inhibition to cause ovarian cancer cell cytotoxicity”. Investigated the potential synergistic effects of CQ + NHEJi and CQ + NHEJi + LBH. Manuscript attempted to build on the previous findings of the group and others. Authors need to address several major and minor concerns outlined below.

Major points:

  • Authors must include colony forming assay for evaluation of cell survival.

We measured cell survival by MTT and by apoptosis assays. These last measures cell apoptosis and necrosis by staining the cells with Annexin V and propidium iodide, so colony forming assays might not be necessary. However, following the reviewer´s recommendation we have performed colony forming assays in the A2780 cell line. The editor gave us 10 days to reply to the reviewers, so we chose this cell line because it is the OC cell line with the highest proliferation rate. The other cell lines would require several weeks to form colonies. We found that A2780 form colonies in the absence of treatment and in the presence of CQ or DNA repair inhibitors (lower size). Cells treated with double combinations hardly formed colonies, as expected. This data has been added as a new supplementary figure (Figure S7).  

  • Non-cancer cell line must be included as an important control for all the assays.

Following the reviewer´s recommendation we have analyzed the effect of the drugs under study in the human embryonic kidney HEK293T cells. We also found synergistic effects in both, double and triple combinations, which is not surprising since they have a very high proliferation rate, even higher than OC cell lines. It is known that cells with high rates of proliferation are more susceptible to DNA damage (doi: 10.1016/j.molonc.2011.07.006) and exhibit high levels of reactive oxygen species. This last can result from increased metabolic activity since ATP synthesis produces ROS during normal oxygen metabolism (doi:10.1007/s11010-019-03667-9).  Several works have combined genotoxic agents with DNA repair inhibitors in vivo and have found cytotoxic effects in tumor cells and lower toxicity in normal cells. Therefore, CQ could drive DNA damage and oxidative pressure to toxic levels selectively or preferentially in tumor cells. The advantages of using CQ together with NHEJ inhibitors and/or HR inhibitors need to be evaluated in in vivo experiments. Depending on the results, encapsulation or targeted delivery of the drugs could be also explored to recapitulate the efficacy of the combinations shown in this work.

Author mentioned that “….in agreement with our results, in vivo xenografts experiments using this cell line revealed that treatment with SCR7 resulted in regression of tumors with no obvious adverse effects” but references and/or data to support this claim are not provided.

The requested reference has been added.

  • Figure 2: For panel C, authors must include error bars, statistics and number of biological replicates. Based on the description provided in the figure legend, it seems that data presented are from only one experiment, which is not acceptable.

Statistics have been incorporated to figure 2C and also the following sentence: “A minimum of 50 cells per condition were analyzed in three independent experiments Error bars represent the SD (***p<0.001; **p<0.01; p<0.05, compared to CQ-treated cells 48 h after CQ removal).

      Figure 3: For panel C, authors must provide statistics and number of biological replicates.

We have provided a better explanation of the statistics and number of replicates (lines 199-204)

  • Figure 4: Authors must include error bars, statistics and number of biological replicates.
  • Figure 6: Authors must include error bars, statistics and number of biological replicates

In this work, CompuSyn software was employed (http://www.combosyn.com) which is based on the Chou-Talalay theory (doi: 10.1158/0008-5472.CAN-09-1947). In this study, Chou T.C indicates that synergism follows a physicochemical mass-action law, not a statistical issue. For this reason, it is recommended to determined synergism by combination indices’ calculation (based on different drug doses at constant ratios), not with p-values. Indeed, a combined effect greater than individual effect does not necessarily indicate synergism and can present a significative p-value. In this case, it can be a result of an additive effect.

Chou-Talalay model can be evaluated by an r value, that indicates the fitting of the data to the mass-action law (doi: 10.1016/j.crphar.2022.100110; doi: 10.1124/pr.58.3.10). In our case, r values were between 0.85 and 0.99 in all the combinations assayed which indicates a confident fitting of the data to the mass-action law.

For all these reasons, we have not included statistics in Figure 4 and 6.

Minor point:

  • There are several typos throughout the manuscript, authors should correct all of them. Few are listed here;

Page 2, line 71, …. proteinWe

Page 2, line 90, .. extrachromosomal

Page 2, line 91,… measuring cell abil-ity

We thank the reviewer; we have revised the manuscript and correct several typos.

Round 2

Reviewer 1 Report

Authors have taken good efforts to address the concerns and revise the manuscript. All questions have been answered and changes have been incorporated to the modified version.

-Would encourage the authors to add the additional references indicated in the response summary to the current version at the relevant locations. 
-Authors have indicated about making a graphical abstract with all mechanisms in response to question 4. However, for some reason, it could not be located.
The manuscript should be good for publication if these comments can also be addressed which should be very straight-forward. 

Author Response

Authors have taken good efforts to address the concerns and revise the manuscript. All questions have been answered and changes have been incorporated to the modified version.

-Would encourage the authors to add the additional references indicated in the response summary to the current version at the relevant locations. 

References indicated in the response summary have been added (lines 89-97) Thanks.

-Authors have indicated about making a graphical abstract with all mechanisms in response to question 4. However, for some reason, it could not be located.
The manuscript should be good for publication if these comments can also be addressed which should be very straight-forward. 

Graphical abstract was uploaded in the system, but maybe you couldn´t access to it.  We will upload a PDF file in the new review report.

Reviewer 3 Report

Dr Herrero and her colleague have nicely responded to almost all my concerns. However, before publication, I’d like to suggest expressing the discussion/explanation they made in the Author’s Notes (the major point 1 and minor point 10) in the paper. I also found the minor point 3 not being corrected yet. 

Author Response

Dr Herrero and her colleague have nicely responded to almost all my concerns. However, before publication, I’d like to suggest expressing the discussion/explanation they made in the Author’s Notes (the major point 1 and minor point 10) in the paper. I also found the minor point 3 not being corrected yet.

Following the reviewer´s recommendation we have included minor point 10 and major point 1 in the paper (lines 346-352 and lines 357-365, respectively).

Regarding minor point 3, the words ex-trachromosomal and abil-ity were corrected the the right forms, extrachromosomal and ability, once changes in the manuscritp (with the track changes) were accepted.

Reviewer 4 Report

The authors have addressed my previous comments to a satisfactory level. I do recommend, that authors should make attempt to validate their MTT data with colony forming assay. 

Author Response

The authors have addressed my previous comments to a satisfactory level. I do recommend, that authors should make attempt to validate their MTT data with colony forming assay. 

Thank you for the positive comments and for the recommendation, in our next experiments we will validate MTT data with colony forming assay, as suggested. This time it wasn´t possible, since we were told to upload the revised file within 3 days.